# A Discrete-Time Extended Kalman Filter Approach Tailored for Multibody Models: State-Input Estimation

**DOI:** 10.3390/s21134495

**Published:** 2021-06-30

**Authors:** Rocco Adduci, Martijn Vermaut, Frank Naets, Jan Croes, Wim Desmet

**Affiliations:** 1LMSD Research Group, Mechanical Engineering Department, KU Leuven University, 3000 Leuven, Belgium; martijn.vermaut@kuleuven.be (M.V.); frank.naets@kuleuven.be (F.N.); jan.croes@kuleuven.be (J.C.); wim.desmet@kuleuven.be (W.D.); 2DMMS Core Labs, Flanders Make, 3001 Heverlee, Belgium

**Keywords:** multibody dynamics, Kalman filtering, coupled states-inputs estimation, virtual sensors, slider-crank mechanism

## Abstract

Model-based force estimation is an emerging methodology in the mechatronic community given the possibility to exploit physically inspired high-fidelity models in tandem with ready-to-use cheap sensors. In this work, an inverse input load identification methodology is presented combining high-fidelity multibody models with a Kalman filter-based estimator and providing the means for an accurate and computationally efficient state-input estimation strategy. A particular challenge addressed in this work is the handling of the redundant state-description encountered in common multibody model descriptions. A novel linearization framework is proposed on the time-discretized equations in order to extract the required system model matrices for the Kalman filter. The presented framework is experimentally validated on a slider-crank mechanism. The nonlinear kinematics and dynamics are well represented through a rigid multibody model with lumped flexibilities to account for localized interaction phenomena among bodies. The proposed methodology is validated estimating the input torque delivered by a driver electro-motor together with the system states and comparing the experimental data with the estimated quantities. The results show the stability and accuracy of the estimation framework by only employing the angular motor velocity, measured by the motor encoder sensor and available in most of the commercial electro-motors.

## 1. Introduction

In mechatronic systems, operational forces and moments are essential quantities in the different stages of the development cycle and strongly impact the design, durability, diagnostic, prognostic, maintenance, and advanced control strategies [1]. However, forces and moments are also difficult, even impossible, quantities to measure. This is due to high force sensor costs and the geometrical constraints (space limitations) that would make the sensor integration impossible without influencing the overall system design and behavior.

In past decades, different test-driven and model-based inverse force methods have been presented in the literature to overcome these limitations. Initially, the challenge of inverse load identification was tackled in offline test-based strategies. One of the most commonly used technique for mechanical applications relates to the field of Transfer Path Analysis (TPA) [2].

TPA summarizes the family of test-based methodologies to study the vibration transmission in mechanical systems where the Matrix Inversion and Mount Stiffness approaches are the most commonly used to estimate inputs and transmitted forces respectively. Despite the wide variety of methods and extensive use in the industrial world, TPA still remains quite expensive from an experimental point of view in terms of preparation and execution time.

The growing computational power of modern computers opened new opportunities to exploit numerical model-based methods. These models can be exploited in virtual sensing approaches [3] which enable the exploitation of low cost, accessible and non-collocated measurements together with first-principle/physically inspired models to obtain state and input estimates.

State estimation techniques such as the Kalman Filter (KF) methods allow the joint estimation of unknown inputs and model states [4] in an efficient manner. By regularly feeding back the measurements on a physical asset, KF techniques enable the compensation of drift in the model while reducing the noise from the direct measurements.

Multibody (MB) modeling approaches are regularly used in the literature and industry [5,6,7] for full scale rigid and flexible mechanical systems where conventional Finite Element (FE)-based methods would be unnecessarily expensive. The MB methods establish a good trade-off between model fidelity and computational cost. Moreover, MB models disclose 3D system-level information, enabling dynamic interaction phenomena among bodies due to distributed and/or localized flexibilities.

However, the link between MB models and estimation algorithms is nontrivial since most estimators require an ordinary differential state-space representation of the system dynamics. Instead, the MB model dynamics, depicted by the Equations of Motion (EOMs), are generally described by a set of Implicit-Differential Algebraic Equations (I-DAEs) that makes the state-space representation difficult to be met. On the other hand, Explicit-Ordinary Differential Equations (E-ODEs) are well suited for a state-space representation but specific MB formulations should be employed to obtain this structure (e.g., [8]), otherwise, dedicated manipulations of the EOMs are demanded to achieve an ODE form.

In [9] the Matrix-R method was proposed to eliminate the constraint equations of the MB model reducing the initial EOMs to an ODE form in independent coordinates. The aim of this work was to combine an extended KF estimator with detailed MB models to obtain an automotive real-time observer. Despite the high accuracy of the estimated quantities, the real-time target was not achieved due to the costly solver iterations.

Similarly in [10,11,12] the Matrix-R method was used to deal with the DAE structure of the EOMs. Here, different KF estimators are compared in terms of accuracy and performance on a rigid 4 and 5-bar linkage mechanisms.

Alternatively, in [13] a kinematic state observer is presented. It combines the constrained kinematic MB equations with nonlinear estimators. Here, the dynamic equations of the MB system are not considered therefore leading to an estimation which is less sensitive to input and mechanical (properties) uncertainties. Moreover the system accelerations are treated as random walk models and augmented to the kinematic discrete state vector that imply the use of a more extensive number of measurement sensors.

In [14,15] an interesting approach based on the combination of deep learning and MB dynamics information was proposed to achieve this transformation. It allows reducing a generic MB model to minimal coordinates allowing the description of the EOMs through E-ODEs while not requiring a specific formulation or access to the constraint equations. However, the methodology depends on a reference numerical simulation as training data which must cover the mechanism workspace; moreover only rigid MB systems can be tackled by the technique.

In [16] an Augmented Discrete Extended Kalman filter (ADE-KF) approach tailored for flexible MB models to construct a state-input estimator is presented.The methodology demonstrates the advantages of using analytical expressions to cover the necessary linearized and explicit EOMs. However, this approach relies on the use of a penalty constraint formulation to achieve E-ODE type of equations. This leads to a relatively poorly conditioned problem and introducing additional model parameters, namely the penalty factors, in comparison to a Lagrange-multiplier approach.

In this work, a generalization of the methodology described in [16] is presented which is compatible with a Lagrange multiplier approach for the constraint equations.

The proposed methodology starts from a novel linearization approach of the EOMs that includes the algebraic variables (Lagrange multipliers) to the system states. Consequently, the resulting unconstrained discrete-time state-space model is employed in a constraint KF scheme where the kinematic constraints are enforced trough the augmented measurement equations, therefore eliminating the effort of selecting effective penalty factors.

The scientific contribution is structured as follows: in Section 2 a general overview of the governing EOMs of the MB system dynamics is given; in Section 3 the implicit constrained EOMs are linearized and made explicit through a first order Taylor expansion; in Section 4 the system and measurement Kalman filter equations are introduced. Here, the system and measurement matrices are analytically assembled thanks to the use of the in-house Multibody Research Code (MBRC) [17]; finally, in Section 5 the methodology is validated on an industrial relevant application comparing the estimated quantities with the experimentally measured one.

## 2. Multi-Body Model and Time-Disretization

This section summarizes the derivation of the EOMs of flexible multibody systems, as they will be employed in this work.

### 2.1. The Multibody Equations of Motion

The mathematical description of the system dynamics can be derived by means of the Lagrange’s equations for constrained mechanical systems [18]:(1)$$\frac{d}{dt}\left(\frac{\partial L(\dot{q},q,\lambda )}{\partial \left[\dot{q},\dot{\lambda }\right]}\right)-\frac{\partial L(\dot{q},q,\lambda )}{\partial \left[q,\lambda \right]}=u^{e}\left(q,u\right),$$
with the Lagrangian defined as:(2)$$L=T-V-\phi {\left(q\right)}^{T}\lambda .$$

L represents the Lagrangian functional, T the kinetic energy, V the potential energy, $\phi {\left(q\right)}^{T}\lambda$ the constraint contribution with the Lagrange multipliers λ, and $u^{e}$ is the vector of the external actions. MB models describe the dynamics of several rigid and/or flexible interacting bodies linked together through the definition of kinematic joints which are mathematically represented by the constraint equations ϕ(q) while J=∂ϕ(q)/∂q represent the Jacobian of the constraint equations. $q\in R^{n_{q}}$ is the generalized coordinates vector, $\lambda \in R^{n_{\lambda }}$ is known as Lagrange multipliers and $u\in R^{n_{u}}$ is the input vector. Through the definition of the assembled body coordinates and the motion parametrization Equation (Equation 1) can be written in a residual form as a fully implicit real-valued non-linear function:(3)$$g(\ddot{q},\dot{q},q,\lambda ,u)=0.$$

### 2.2. The Differential-Algebraic form of the EOMs

A set of natural coordinates $q_{n}\in R^{n_{q_{n}}}$ was proposed in [18], where redundant degrees of freedom are employed to define the system coordinates of the assembled bodies. Moreover, including the motion parameterization employed in the Flexible Natural Coordinate Formulation (FNCF) [19] allows deriving a constant singular mass matrix $M\in R^{n_{q_{n}}\times n_{q_{n}}}$. Assuming this formulation, Equation (Equation 3) can be written in the so-called index-3 form:(4)$$\begin{array}{c} \left\{\begin{array}{c} g_{1}=M_{n}{\ddot{q}}_{n}+f_{nl}\left({\dot{q}}_{n},q_{n},u\right)+J^{T}\lambda =0_{q}\\ g_{2}=\phi \left(q_{n}\right)=0_{\lambda } \end{array}\right.. \end{array}$$

Here, $f_{nl}\in R^{n_{q_{n}}}$ is the non-linear generalized force vector expressed as:(5)$$f_{nl}\left({\dot{q}}_{n},q_{n},u\right)=f_{v}\left({\dot{q}}_{n},q_{n}\right)+f_{int}\left({\dot{q}}_{n},q_{n}\right)+f_{ext}\left({\dot{q}}_{n},q_{n},u\right).$$

$f_{v}$ represents the quadratic velocity vector related to the gyroscopic forces of the bodies, which is zero for FNCF formulation. $f_{int}$ is the internal force vectors which accounts for the elastic energy stored by deformable bodies and if rigid bodies are assumed $f_{int}$ vanishes; $f_{ext}$ is the external force vector and can be spilt in the sum of two contributions, the interaction forces among bodies $f_{b}$ (i.e., contact and friction forces) and the input forces $f_{u}$. They can be summarized as follows:(6)$$f_{ext}\left({\dot{q}}_{n},q_{n},u\right)=f_{b}\left({\dot{q}}_{n},q_{n}\right)+f_{u}\left(q_{n},u\right).$$

Here, $f_{u}$ can be written as $f_{u}\left(q_{n},u\right)=U_{t}\left(q_{n}\right)u$, where $U_{t}$ is tangent input matrix defined as:(7)$$U_{t}=\frac{\partial f_{u}}{\partial u}.$$

Due to the structure of the EOM, Equation (Equation 4), for the FNCF formulation, derivatives can be more readily obtained than for may alternative flexible multibody formulations. Therefore, the above mentioned coordinates definition and motion parameterization will be considered in this work. For the sake of brevity, we omit the subscript *n* referred to the natural coordinate formulation for the remainder of this manuscript.

Despite the computational advantages of the above mentioned MB approach, the methodologies that will be introduced in the next sections can be easily extended to alternative MB formulation, such as the floating-frame of reference component mode synthesis approach or the generalized component mode synthesis [5,20].

The I-DAEs form of Equation (Equation 4) are generally not suitable for estimation algorithms such as the Kalman Filter family, since these have been designed to handle E-ODEs type of equations.

In the next section, we present a new methodology to directly linearize the I-DAEs starting from its discrete form but without employing any explicit constraint elimination technique.

### 2.3. EOMs: The Discrete Index-3 Form

To transform the second order differential equation into first-order differential equations, it is common to introduce the velocity variable $\dot{q}=v$, allowing to write Equation (Equation 4) in a first-order form as:(8)$$\begin{array}{cc} \; & g\left(\dot{v},v,\dot{q},q,\lambda ,u\right)=\left\{\begin{array}{c} g_{1}=v-\dot{q}=0_{v}\\ g_{2}=M\left(q\right)\dot{v}+f_{nl}\left(v,q,u\right)+J^{T}\lambda =0_{q}\\ g_{3}=\phi \left(q\right)=0_{\lambda } \end{array}\right.. \end{array}$$

These equations represent a system of constrained-DAEs of index 3 [21]. Numerical differentiation is generally employed [22] to convert Equation (Equation 8) into a discrete form.

In this work, a first order, constant time-step Backward Differentiation Formula (BDF-1), also known as Backward Euler, is employed. This choice does not imply that other differentiation schemes cannot be applied to discretize the EOMs in time. However, the time-discretization must be consistent with the defined estimation sampling and particular attention should be paid to choosing the differentiation schemes (e.g., forward or backward) because it influences the achievement of the discrete-time E-ODE form of the EOMs, as will be discussed in Section 3.

The single step method BDF-1 can be written as
(9)$$\left\{\begin{array}{c} {\dot{v}}_{k+1}=\frac{1}{h}\left(v_{k+1}-v_{k}\right)\\ v_{k+1}=\frac{1}{h}\left(q_{k+1}-q_{k}\right) \end{array}\right.,$$
where h represents the constant time step size and k∈Z the k−th time instance. By substituting Equation (Equation 9) into Equation (Equation 8) the discrete-time EOMs $g_{d}$ are obtained:(10)$$\left\{\begin{array}{c} g_{d1}=v_{k+1}-\frac{1}{h}\left(q_{k+1}-q_{k}\right)=0_{v}\\ g_{d2}=M\left(q_{k+1}\right)\frac{v_{k+1}-v_{k}}{h}+f_{nl}\left(q_{k+1},q_{k},u_{k+1}\right)+J^{T}\left(q_{k+1}\right)\lambda_{k+1}=0_{q}\\ g_{d3}=\phi \left(q_{k+1}\right)=0_{\lambda } \end{array}\right..$$

This can be summarized as
(11)$$g_{d}\left(v_{k+1},q_{k+1},v_{k},q_{k},\lambda_{k+1},u_{k+1}\right)=0.$$

Assuming the generalized coordinates $q_{k}$ and velocities $v_{k}$ at the time instance *k* to be known and given the input $u_{k+1}$ at the time instance k+1, Equation (Equation 10) is solved for $q_{k+1}$ and $\lambda_{k+1}$ by substituting $g_{d1}$ in $g_{d2}$, and applying a Newton-Raphson-based iterative algorithm with iteration gradient $J_{NR}$:(12)$$J_{NR}=\left[\begin{array}{ccc} K_{t}+\beta C_{t}+\gamma M_{t} & \; & J^{T}\\ J & \; & 0_{\lambda ,\lambda } \end{array}\right].$$

Here, $K_{t}$, $C_{t}$ and $M_{t}$ are the tangent stiffness, damping and mass matrices obtained from the partial derivatives of the continuous $g_{2}$ equations in Equation (Equation 8) evaluated at time step k+1:(13)$$\begin{array}{ccccc} K_{t}=\frac{\partial g_{2}}{\partial q}{|}_{k+1}; & \; & C_{t}=\frac{\partial g_{2}}{\partial v}{|}_{k+1}; & \; & M_{t}=\frac{\partial g_{2}}{\partial \dot{v}}{|}_{k+1}, \end{array}$$
and β and γ are matrix coefficients function of the defined integration rule, which are given for the BDF-1 scheme by:(14)$$\begin{array}{ccc} \beta =\frac{\partial v_{k+1}}{\partial q_{k+1}}=\frac{\partial {\dot{v}}_{k+1}}{\partial v_{k+1}}=\frac{1}{h}I_{q}; & \; & \gamma =\frac{\partial {\dot{v}}_{k+1}}{\partial q_{k+1}}=\frac{1}{h^{2}}I_{q}; \end{array}$$
(15)$$\begin{array}{ccc} \frac{\partial v_{k+1}}{\partial q_{k}}=\frac{\partial {\dot{v}}_{k+1}}{\partial v_{k}}=-\beta ; & \; & \frac{\partial {\dot{v}}_{k+1}}{\partial q_{k}}=-\gamma . \end{array}$$

This time integration scheme will be exploited in the following section to set up a suitable solver structure for use in an ODE-base estimation framework.

## 3. An Explicit Linearized Approximation for Use of the Multibody Model in State-Estimation

The aim of this work is to combine MB models with a Kalman filter-based estimator to concurrently estimate the system states and unknown forces of a mechanism. These presented estimators, as will be discussed in Section 4, require the linearized time-discrete model matrices. In this section, a new approach to linearize the EOMs of a MB model starting from an ID-DAE form is presented. Subsequently, the linearized system matrices are analytically assembled.

By introducing the state vector *x*$\in R^{n_{x}}$ for a time step *k*
(16)$$x_{k}={\left[\begin{array}{ccc} v_{k} & q_{k} & \lambda_{k} \end{array}\right]}^{T},$$
the ID-DAE form of the EOMs in Equation (Equation 11) can be re-written as:(17)$$g_{d}\left(x_{k+1},x_{k},u_{k+1}\right)=0_{x}.$$

In this work we assume the function $g_{d}$ of Equation (Equation 17) to be continuously differentiable in all its variables. Therefore, an explicit discrete function $f_{d}$ locally exists by applying the implicit function theorem. Through a first order Taylor expansion, $g_{d}$ can be approximated as
(18)$$\begin{array}{c} g_{d}^{0}+\frac{\partial g_{d}}{\partial x_{k+1}}{|}_{0}\left(x_{k+1}-x_{k+1}^{0}\right)+\frac{\partial g_{d}}{\partial x_{k}}{|}_{0}\left(x_{k}-x_{k}^{0}\right)+\ldots \\ \frac{\partial g_{d}}{\partial u_{k+1}}{|}_{0}\left(u_{k+1}-u_{k+1}^{0}\right)+O\left(x_{k+1},x_{k},u_{k+1}\right)=0_{x}; \end{array}$$
where ($x_{k+1}^{0}$,$x_{k}^{0}$,$u_{k+1}^{0}$) represents the linearization set point while $g_{d}^{0}=g_{d}(x_{k+1}^{0}$,$x_{k}^{0}$,$u_{k+1}^{0})$ for convenience of notation. Manipulating Equation (Equation 18) and neglecting the higher order terms, it can be made explicit as
(19)$$x_{k+1}=x_{k+1}^{0}-{\left[\frac{\partial g_{d}}{\partial x_{k+1}}{|}_{0}\right]}^{-1}\left(g_{d}^{0}+\frac{\partial g_{d}}{\partial x_{k}}{|}_{0}\left(x_{k}-x_{k}^{0}\right)+\frac{\partial g_{d}}{\partial u_{k+1}}{|}_{0}\left(u_{k+1}-u_{k+1}^{0}\right)\right).$$

In compact form Equation (Equation 19) becomes:(20)$$x_{k+1}=f_{d}\left(x_{k},u_{k+1}\right)=x_{k+1}^{0}+f_{d}^{0}\left(x_{k},u_{k+1}\right)+A_{k+1}^{0}\left(x_{k}-x_{k}^{0}\right)+B_{k+1}^{0}\left(u_{k+1}-u_{k+1}^{0}\right).$$

$A_{k+1}^{0}=\frac{\partial f_{d}}{\partial x}{|}_{k+1}^{0}$ and $B_{k+1}^{0}=\frac{\partial f_{d}}{\partial u}{|}_{k+1}^{0}$ are the linearized system and input matrices around the linearization set point.

Starting from Equation (Equation 19) and differentiating Equation (Equation 10), the Jacobians for the backward Euler implicit time-integrator can be computed as:(21)$$G_{x_{k+1}}=\frac{\partial g_{d}}{\partial x_{k+1}}=\left[\begin{array}{ccc} \frac{\partial g_{d1}}{\partial v_{k+1}} & \frac{\partial g_{d1}}{\partial q_{k+1}} & \frac{\partial g_{d1}}{\partial \lambda_{k+1}}\\ \frac{\partial g_{d1}}{\partial v_{k+1}} & \frac{\partial g_{d2}}{\partial q_{k+1}} & \frac{\partial g_{d2}}{\partial \lambda_{k+1}}\\ \frac{\partial g_{d3}}{\partial v_{k+1}} & \frac{\partial g_{d3}}{\partial q_{k+1}} & \frac{\partial g_{d3}}{\partial \lambda_{k+1}} \end{array}\right]=\left[\begin{array}{ccc} I_{v} & -\beta & 0_{v,\lambda }\\ 0_{q,v} & \gamma M_{t}+\beta C_{t}+K_{t} & J^{T}\\ 0_{\lambda ,v} & J & 0_{\lambda ,\lambda } \end{array}\right];$$
(22)$$G_{x_{k}}=\frac{\partial g_{d}}{\partial x_{k}}=\left[\begin{array}{ccc} \frac{\partial g_{d1}}{\partial v_{k}} & \frac{\partial g_{d1}}{\partial q_{k}} & \frac{\partial g_{d1}}{\partial \lambda_{k}}\\ \frac{\partial g_{d2}}{\partial v_{k}} & \frac{\partial g_{d2}}{\partial q_{k}} & \frac{\partial g_{d2}}{\partial \lambda_{k}}\\ \frac{\partial g_{d3}}{\partial v_{k}} & \frac{\partial g_{d3}}{\partial q_{k}} & \frac{\partial g_{d3}}{\partial \lambda_{k}} \end{array}\right]=\left[\begin{array}{ccc} 0_{v,v} & \beta & 0_{v,\lambda }\\ -\beta M_{t} & -\gamma M_{t}-\beta C_{t} & 0_{q,\lambda }\\ 0_{\lambda ,v} & 0_{\lambda ,q} & 0_{\lambda ,\lambda } \end{array}\right];$$
(23)$$G_{u_{k+1}}=\frac{\partial g_{d}}{\partial u_{k+1}}=\left[\begin{array}{c} \frac{\partial g_{d1}}{\partial u_{k+1}}\\ \frac{\partial g_{d2}}{\partial u_{k+1}}\\ \frac{\partial g_{d3}}{\partial u_{k+1}} \end{array}\right]=\left[\begin{array}{c} 0_{v,u}\\ U_{t}\\ 0_{\lambda ,u} \end{array}\right],$$
where $U_{t}$ represents the tangent input matrix of Equation (Equation 7).

The linearized system and input matrices can be computed for any working point at the time step k+1 as:(24)$$A_{k+1}=-G_{x_{k+1}}^{-1}G_{x_{k}};$$
(25)$$B_{k+1}=-G_{x_{k+1}}^{-1}G_{u_{k+1}}.$$

These equations enable the evaluation of the linearized time-discretized explicit description of the EOMs suitable for estimation methods such as the extended Kalman filter. Their deployment of this estimation scheme is discussed in the next section.

It is important to note that the invertibility of the matrix $G_{x_{k+1}}$ is guaranteed thanks to the choice of BDF-1 differentiation scheme and that in contrast, it would not be possible. For completeness, the limitations of a forward differentiation scheme is demonstrated in the Appendix A.

## 4. State-Input Estimation for MB Models

The augmented discrete extended Kalman filter (ADE-KF) tailored for MB models is discussed in this section. In this section we discuss all the required components to set up a Kalman filter for assimilating the different multibody states and unknown inputs. More in details, in Section 4.1, the general form of the discrete-time system and measurement model equations are summarized. In Section 4.2, the measurement equations are augmented with the constraint equations ϕ, to deal with the constrained nature of the MB EOMs, leading to a constrained estimation problem. Moreover, in Section 4.3 the adopted approach to compute the linearized measurement matrices *C* and *D* are presented since they are required in the estimation framework and not directly available. Finally, in Section 4.4 the ADE-KF is assembled to deal with the estimation of states and unknown inputs, and the Kalman filter steps tailored for MB models are reported.

### 4.1. Model and Measurement Equations with Uncertainty

The system of EOMs in Equation (Equation 4) described in the previous section are assumed as deterministic. However, in practice, various noise sources will disturb the behavior of the system. For the dynamic model equations, the process noise vector $\nu_{x,k+1}^{*}$ influences the response:(26)$$g_{d}\left(x_{k+1},x_{k},u_{k+1}\right)={\tilde{\nu }}_{x,k+1},$$
where ${\tilde{\nu }}_{x,k+1}$ is associated with the noise term $\nu_{x,k+1}$ described by the following equation:(27)$$x_{k+1}=x_{k+1}^{*}+\nu_{x,k+1}.$$

Here, $x_{k+1}^{*}$ is the deterministically varying state vector while $\nu_{x,k+1}$ is a zero-mean noise term with a (possibly time-varying) covariance $Q_{k}$.

The estimation framework relies on combining the prior model information with measurements $y\in R^{n_{y}}$ on the physical process. To compare the prediction of the model with these measurements, a set of measurement equations *h* are necessary which are affected by measurement noise $\nu_{y,k+1}$:(28)$$y=h\left({\dot{v}}_{k+1},x_{k+1},u_{k+1}\right)+\nu_{y,k+1},$$
where again $\nu_{y,k+1}$ is assumed to be zero-mean with a (time-varying) covariance $R_{k}$. Similar to the model measurement equations *y*, the concept of virtual sensor (VS) $y_{vs}\in R^{n_{y_{vs}}}$ is introduced:(29)$$y_{vs}=h_{vs}\left({\dot{v}}_{k+1},x_{k+1},u_{k+1}\right)+\nu_{vs}.$$

A virtual sensor represents model-based information; more specifically, it is a physical quantity that can be expressed as a function of the system variables, such as joint forces, body motion trajectories (e.g., linear and angular positions, velocities, and accelerations) and body strains/stresses. A VS equation can be treated similarly to a measurement equation but evaluated after the a posteriori Kalman filter step and hence to exploited in the estimation process itself.

### 4.2. The Augmented Constraint Measurement Equations

In [23] different approaches have been proposed to deal with constrained state estimation, although the authors deem it impossible to give a general guideline for constrained filtering because each individual problem is unique and dependent on how the constraints are handled.

In this work, the combination of hard and soft constraint methods is considered [23], where the constraint measurement equations are implemented in the Kalman filter scheme by augmenting the measurement equations *y* of Equation (Equation 28) with the constraint equations:(30)$$\tilde{y}=\left\{\begin{array}{c} h\left({\dot{v}}_{k+1},x_{k+1},u_{k+1}\right)+\nu_{y,k+1}\\ \phi \left(q_{k+1}\right)+\nu_{\phi ,k+1} \end{array}\right..$$

Here, $\nu_{\phi ,k+1}$ is a small zero-mean noise term added to the idealized constraint equations $\phi (q_{k+1})$ if small constraints violation is allowed (soft constraints method). In the case $\nu_{\phi ,k+1}$ is assumed to be zero, the constraint equations are treated as perfect measurements (hard constraint method).

### 4.3. An Efficient Strategy for the Measurement Sensitivities Computation

To evaluate the EKF equations at each filter step, the linearized measurement sensitivity matrices $C_{k+1}$ and $D_{k+1}$ are required and they are defined as:(31)$$C_{k+1}=\frac{\partial \tilde{y}}{\partial x}{|}_{k+1}={\left[\begin{array}{c} \frac{dy}{dx}\\ \frac{\partial \phi }{\partial x} \end{array}\right]}_{k+1}={\left[\begin{array}{ccc} \frac{dy}{dv} & \frac{dy}{dq} & \frac{dy}{d\lambda }\\ 0_{\lambda ,v} & J & 0 \end{array}\right]}_{k+1};$$
(32)$$D_{k+1}=\frac{\partial \tilde{y}}{\partial u}{|}_{k+1}={\left[\begin{array}{c} \frac{dy}{du}\\ 0_{\lambda ,u} \end{array}\right]}_{k+1}.$$

Generally, the measurement equations can be expressed as a function of the generalized accelerations, velocities, positions, Lagrange multipliers, and inputs, Equation (Equation 30).

However, due to the state-space description presented in Section 3, the explicit relationship of the sensor equations with respect to the generalized accelerations is lost. More in particular, when considering acceleration sensors, their sensitivity with respect to the generalized accelerations is non zero, thus it must be indirectly included. Acceleration sensors are really common in the mechanical engineering community due to their non-invasive and simple installation requirements. Within the estimation framework, they can be used either in form of measurements or in form of VSs.

Similarly, the sensitivity of the measurement equations with respect to the system inputs are not always available or nontrivial to obtain.

Therefore, an approach to explicitly obtain these dependencies is here derived by means of the acceleration level constraints.

Introducing the auxiliary variables $z\in R^{3n_{q}+n_{\lambda }}$
(33)$$z={\left[\begin{array}{cccc} {\ddot{q}}^{T} & {\dot{q}}^{T} & q^{T} & \lambda^{T} \end{array}\right]}^{T}={\left[\begin{array}{cccc} {\dot{v}}^{T} & v^{T} & q^{T} & \lambda^{T} \end{array}\right]}^{T},$$
for a generic measurement sensor *y*, the sensitivities $\frac{\partial y}{\partial z}$ with respect to the auxiliary variable *z* can be analytically computed starting from their fundamental equations.
(34)$$\frac{\partial y}{\partial z}=\left[\begin{array}{cccc} \frac{\partial y}{\partial \dot{v}} & \frac{\partial y}{\partial v} & \frac{\partial y}{\partial q} & \frac{\partial y}{\partial \lambda } \end{array}\right].$$

Applying the chain rule, the non-augmented linear measurement matrix $\frac{dy}{dx}$ for a generic time step can be written as
(35)$$\frac{dy}{dx}=\frac{\partial y}{\partial z}\frac{dz}{dx}.$$

And the full derivative $\frac{dz}{dx}$ can be expressed as
(36)$$\frac{dz}{dx}=\left[\begin{array}{ccc} \frac{\partial \dot{v}}{\partial v} & \frac{\partial \dot{v}}{\partial q} & \frac{\partial \dot{v}}{\partial \lambda }\\ \frac{\partial v}{\partial v} & \frac{\partial v}{\partial q} & \frac{\partial v}{\partial \lambda }\\ \frac{\partial q}{\partial v} & \frac{\partial q}{\partial q} & \frac{\partial q}{\partial \lambda }\\ \frac{\partial \lambda }{\partial v} & \frac{\partial \lambda }{\partial q} & \frac{\partial \lambda }{\partial \lambda } \end{array}\right]=\left[\begin{array}{ccc} \frac{\partial \dot{v}}{\partial v} & \frac{\partial \dot{v}}{\partial q} & \frac{\partial \dot{v}}{\partial \lambda }\\ I_{v} & 0_{v,q} & 0_{v,\lambda }\\ 0_{q,v} & I_{q} & 0_{q,\lambda }\\ \frac{\partial \lambda }{\partial v} & \frac{\partial \lambda }{\partial q} & I_{\lambda } \end{array}\right].$$

To analytically compute the unknown terms of Equation (Equation 36), we introduce the acceleration level constraints $\ddot{\phi }$ defined as the second time derivative of the position level constraints. Starting from the velocity level constraints $\dot{\phi }$:(37)$$\dot{\phi }=\frac{\partial \phi }{\partial q}\frac{dq}{dt}=Jv,$$
the acceleration level constraint equations are obtained as follows:(38)$$\ddot{\phi }=\dot{J}v+J\dot{v}=\sum_{i,j=1}^{n_{q}}v_{i}\frac{\partial^{2}\phi }{\partial \left[q_{i},q_{j}\right]}v_{j}+J\dot{v}=0_{\lambda }.$$

By means of the continuous dynamic equations $g_{2}$ of Equation (Equation 8) and the acceleration level constraint of Equation (Equation 38), a new set of equations $p\in R^{n_{q}+n_{\lambda }}$ can be constructed:(39)$$p\left(z,u\right)=\left\{\begin{array}{c} p_{1}=M\left(q\right)\dot{v}+f_{nl}\left(v,q,u\right)+J^{T}\lambda =0_{q}\\ p_{2}=\sum_{i,j=1}^{n_{q}}v_{i}\frac{\partial^{2}\phi }{\partial \left[q_{i},q_{j}\right]}v_{j}+J\dot{v}=0_{\lambda } \end{array}\right..$$

Equations (Equation 39) represent the governing set of equations that implicitly determines the relation between the different coordinates *v*, *q*, $\dot{v}$ and the Lagrange multipliers λ.

Therefore, the unknown terms of Equation (Equation 36) can be computed as
(40)$$\begin{array}{cc} \; & \left[\begin{array}{ccc} \frac{\partial \dot{v}}{\partial v} & \frac{\partial \dot{v}}{\partial q} & \frac{\partial \dot{v}}{\partial \lambda }\\ \frac{\partial \lambda }{\partial v} & \frac{\partial \lambda }{\partial q} & \frac{\partial \lambda }{\partial \lambda } \end{array}\right]=-{\left[\begin{array}{cc} \frac{\partial p}{\partial \dot{v}} & \frac{\partial p}{\partial \lambda } \end{array}\right]}^{-1}\left[\begin{array}{ccc} \frac{\partial p}{\partial v} & \frac{\partial p}{\partial q} & \frac{\partial p}{\partial \lambda } \end{array}\right]=\\ \; & -{\left[\begin{array}{cc} M(q) & J^{T}\\ J & 0_{\lambda ,\lambda } \end{array}\right]}^{-1}\left[\begin{array}{ccc} C_{t} & K_{t} & J^{T}\\ 2\sum_{j=1}^{n_{v}}\frac{\partial^{2}\phi }{\partial \left[q,q_{j}\right]}v_{j} & \sum_{j=1}^{n_{\dot{v}}}\frac{\partial^{2}\phi }{\partial \left[q,q_{j}\right]}{\dot{v}}_{j} & 0_{\lambda ,\lambda } \end{array}\right], \end{array}$$
where the third order partial derivative of the constraint equations ϕ(q) with respect to the generalized coordinates *q* is zero for FNCF. Another important ingredient to fulfill the Kalman filter requirements is the measurement input matrix $\frac{dy}{du}$ of Equation (Equation 32). However, this matrix is not directly available and it is computed similarly to Equation (Equation 35) but with respect to input *u*:(41)$$\frac{dy}{du}=\frac{\partial y}{\partial z}\frac{dz}{du}+\frac{\partial y}{\partial u}.$$

Here, the full derivative $\frac{dz}{du}$ represents how the vector *z* varies when the input *u* is perturbed obtaining:(42)$$\frac{dz}{du}=\left[\begin{array}{cccc} \frac{\partial \dot{v}}{\partial u}\; & \frac{\partial v}{\partial u}\; & \frac{\partial q}{\partial u}\; & \frac{\partial \lambda }{\partial u} \end{array}\right]=\left[\begin{array}{cccc} \frac{\partial \dot{v}}{\partial u}\; & 0_{v,u}\; & 0_{q,u}\; & \frac{\partial \lambda }{\partial u} \end{array}\right].$$

From Equation (Equation 42) can be seen that only the sensitivities of the acceleration and Lagrange multipliers are non-zero terms. A second order system is fully defined by the position coordinates, velocity coordinates, and time. An external force only influences the force balance of that system and thus the acceleration coordinates and Lagrangian multipliers while only through time-integration the velocity and position coordinates. However, these derivatives are evaluated at a single time instance. Therefore, in Equation (Equation 36), the acceleration coordinates and Lagrangian multipliers depend on the position and velocity coordinates, but not the other way around: the position and velocity coordinates do not depend on the acceleration coordinates, Lagrangian multipliers, or the external forces.

Subsequently, the remaining terms of Equation (Equation 42) can be computed as
(43)$$\left[\begin{array}{c} \frac{\partial \dot{v}}{\partial u}\\ \frac{\partial \lambda }{\partial u} \end{array}\right]=-{\left[\begin{array}{cc} \frac{\partial p}{\partial \dot{v}} & \frac{\partial p}{\partial \lambda } \end{array}\right]}^{-1}\frac{\partial p}{\partial u}=-{\left[\begin{array}{cc} M(q) & J^{T}\\ J & 0_{\lambda ,\lambda } \end{array}\right]}^{-1}\left[\begin{array}{c} U_{t}\\ 0_{\lambda ,u} \end{array}\right].$$

In Figure 1 a schematic summary of the followed steps required for the computation of the system and measurement matrices is reported.

### 4.4. Augmented Discrete Extended Kalman Filter

The goal of this work is to estimate the unknown input forces, often referred to as disturbances, $d\in R^{n_{d}}$, concurrently with the system states *x* through the augmented state vector $\tilde{x}\in R^{n_{x}+n_{d}}$ defined as:(44)$${\tilde{x}}_{k+1}=\left[\begin{array}{c} x_{k+1}\\ d_{k+1} \end{array}\right].$$

As the estimation algorithm also needs a model for the unknown inputs, a random walk model is associated with the unknown input:(45)$$d_{k+1}=d_{k}+\nu_{d,k};$$
with $\nu_{d,k}$ also being zero-mean, random noise with covariance matrix $Q_{d,k}$.

In this work, the random walk model is chosen for the unknown disturbance. This approach is chosen as in general no prior information on the input is assumed to be known. However, if useful information is available, e.g., periodicity of the input, other input models can be employed, as in [24].

The augmented discrete-time state-space system of equations can be written combining Equations (Equation 26) and (Equation 45):(46)$${\tilde{g}}_{d}\left({\tilde{x}}_{k+1},{\tilde{x}}_{k},u_{k+1}\right)=\left\{\begin{array}{c} g_{d}\left(x_{k+1},x_{k},u_{k+1}\right)={\tilde{\nu }}_{x,k}\\ d_{k+1}-d_{k}=\nu_{d,k} \end{array}\right..$$

The linearized system matrix F refers to the augmented system of equations Equation (Equation 46) at the time step k+1 and it is assembled as
(47)$$F=\frac{\partial {\tilde{f}}_{d}}{\partial \tilde{x}}{|}_{k+1}=\left[\begin{array}{cc} A_{k+1} & B_{k+1}^{d}\\ 0_{x,d} & I_{d} \end{array}\right];$$
where ${\tilde{f}}_{d}$ is the augmented explicit ODEs associated with Equation (Equation 46), is obtained from Equation (Equation 24), while $B_{k+1}^{d}$ is the linear disturbance matrix obtained similarly to $B_{k+1}$ in Equation (Equation 25) but with respect to the unknown input *d*.

Subsequently, the linearized augmented measurement matrix H associated with the augmented state vector $\tilde{x}$ is assembled as:(48)$$H=\frac{\partial \tilde{y}}{\partial \tilde{x}}{|}_{k+1}=\left[\begin{array}{cc} C_{k+1} & D_{k+1}^{d} \end{array}\right],$$
where the matrix $D_{k+1}^{d}$ is computed in a similar fashion as for $D_{k+1}$ in Equation (Equation 32) but with respect to the unknown input *d*.

Starting from these model, measurement and tangent matrices, an extended Kalman filter (EKF) procedure can be set up in order to estimate both the states and the unknown input forces. The application of this estimation approach is discussed in the following section.

### 4.5. The Adopted Extended Kalman Filter Scheme

In this work, we employ an extended Kalman filter (EKF) [25] to perform the state-estimation. With the various model and measurement equations, and their respective derivatives, defined for a MB model in the previous sections this EKF can be run efficiently. As all terms have been defined analytically it is not necessary to resort to numerical derivatives, which allows for an effective application of the EKF. Several researchers have shown the applicability of the estimators even for the strongly nonlinear case of MB systems [9,12,26].

In general, we can group the EKF scheme in two main steps: the a-priori and a posteriori steps.

*A-priori step*: Assuming that the augmented states ${\tilde{x}}_{k}^{+}$ at the previous filter step and the input $u_{k+1}$ are known, the a-priori state prediction ${\tilde{x}}_{k+1}^{-}$ and generalized accelerations ${\dot{v}}_{k+1}^{-}$ can be computed solving the ID-DAEs of Equation (Equation 17):
(49)$$g_{d}\left({\tilde{x}}_{k+1}^{-},{\tilde{x}}_{k}^{+},u_{k+1}\right)=0_{x}.$$Knowing the estimated state covariance matrix $P_{k}^{+}$ for the previous timestep, the a-priori covariance at the current time (k+1) step can be approximated from Equation (Equation 47) as
(50)$$P_{k+1}^{-}=FP_{k}^{+}F^{T}+{\tilde{Q}}_{k}.$$
where ${\tilde{Q}}_{k}$ is the process noise matrix of the augmented state-space model.The predicted measurement ${\tilde{y}}_{k+1}^{-}$ can then be evaluated from Equation (Equation 30) as:
(51)$${\tilde{y}}_{k+1}^{-}=\tilde{y}\left({\dot{v}}_{k+1}^{-},x_{k+1}^{-},u_{k+1}\right).$$The Kalman filter gain K allows achieving a desireable trade-off between the confidence in the model and the available measurements, and can be evaluated as:
(52)$$K_{k+1}=P_{k+1}^{-}H^{T}{\left(HP_{k+1}^{-}H^{T}+{\tilde{R}}_{k}\right)}^{-1},$$
where H is obtained from Equation (Equation 48) and ${\tilde{R}}_{k}$ is the measurement covariance matrix of the augmented measurement equations.*A-posteriori step*: When the real measurement $y_{k+1}^{*}$ becomes available together with the predicted measurement ${\tilde{y}}_{k+1}^{-}$, the a posteriori state vector ${\tilde{x}}_{k+1}^{+}$ is obtained as:
(53)$${\tilde{x}}_{k+1}^{+}={\tilde{x}}_{k+1}^{-}+K_{k+1}\left(y_{k+1}^{*}-{\tilde{y}}_{k+1}^{-}\right)$$The inclusion of the actual measurements also affects the posterior covariance matrix $P_{k+1}^{+}$ and can be evaluated as:
(54)$$P_{k+1}^{+}=\left(I_{x}-K_{k+1}H\right)P_{k+1}^{-}.$$

In Figure 2 a block diagram scheme summarizing a generic kth step of the recursive ADE-KF is shown.

With this scheme the entire state-input estimation can be performed for an arbitrary (flexible) multibody model. In the following section we will validate it on an experimental setup.

In practical applications the performance of the estimation scheme will strongly depend on the tuning of the model covariance $\tilde{Q}$ and measurement covariance $\tilde{R}$. Unfortunately this tuning is also highly case dependent, which makes it difficult to set up general guidelines. For the particular validation case considered in this work, a detailed discussion on a possible tuning strategy is presented in the following section.

## 5. Validation: Joint State-Input Estimation

In this section, the presented joint state-input estimation methodology is experimentally validated on a slider-crank mechanism.

### 5.1. The Slider-Crank System

The slider-crank system in Figure 3 is a mechanism widely used in industry to transform a rotational motion in a linear motion as for instance in pick and place operations or in car engines.

Figure 4 represents the multibody model of the experimental setup shown in Figure 3 consisting of 14 bodies: base block (BB), motor housing (MH), motor rotor (MR), motor support (MS), crank (*C*), crank shaft (CSh), crank support (CS), connecting rod (CR), crank-rod bearing ($B_{C-CR}$), slider (*S*), rod-slider bearing ($B_{CR-S}$), slider accelerometer (SA), track rail (TR) and track support (TS).

The mechanical properties of the various bodies are listed in Table 1.

Even though the presented slider-crank mechanism is an academic example it combines several challenging phenomena such as non-linear kinematics and complex slider-track interaction phenomena. The various bodies are connected trough fixed, revolute, spherical, and universal joints to allow the desired kinematics. The slider and track bodies are connected through a contact stiffness $k_{c}$ along the perpendicular directions to the sliding trajectory. The Pacejka friction model [27] defines the friction coefficient μ as a function of the sliding velocity Δv:(55)$$\mu =dsin\left[ctan^{-1}\left(b\Delta v-e\left(b\Delta v-tan^{-1}\left(b\Delta v\right)\right)\right)\right],$$
where the friction force $f_{f}$ is evaluated as
(56)$$f_{f}=\mu f_{n}=\mu \left(k_{c}\delta +c_{c}\dot{\delta }\right),$$
where $f_{n}$ is the resultant normal force acting on the slider and, δ is the local compliance between the slider and track rail interfaces projected onto the normal direction to the sliding direction. The adopted (identified) Pacejka friction model parameters are listed in Table 2.

The system motion is driven by a brushless servomotor “MAC3000”’ with integrated controller MAC00-B4 from JVL (www.jvl.dk, accessed on 1 March 2021). It includes a motor encoder sensor allowing the measurement of the rotation angle and velocity of the motor shaft, indicated by θ and $\dot{\theta }$ respectively.As can be seen in Figure 3 and Figure 4, the slider is equipped with a mono-axial MEMS accelerometer (3711D1FB200G) from PCB (www.pcb.com) to measure its translational acceleration along the sliding direction indicated by $\ddot{Y}$. The entire system is controlled via the motor using a closed-loop PID controller targeting a desired rotational motor velocity while adapting the determined motor torque *T*.

### 5.2. Results

After the setup of the MB model of the slider-crank mechanism, it is embedded into the ADE-KF estimation framework presented in Section 4.

The input torque delivered by the motor is assumed unknown, and jointly estimated with the model states as introduced in Section 4.4. It is assumed that all model uncertainties is dominated by the augmented state, representing the unknown input, while the model is considered perfect.

The a posteriori Kalman filter step is computed using the augmented measurements discussed in Section 4.2. These combine the angular motor velocity $\dot{\theta }$ together with the model constraint equations ϕ. The angular motor velocity is directly available on many mechatronic drives since they are equipped with relatively accurate encoders for control or monitoring purposes.

For the validation of our estimation framework, summarized in Figure 5, we compare the estimates (virtual sensors) of the motor position θ, motor velocity $\dot{\theta }$, slider acceleration $\ddot{Y}$ and motor torque *T* with measurements directly obtained from the experimental setup. Besides the motor encoder, an accelerometer on the slider is employed and motor torques can be directly extracted from the drive unit.

The performed experiments span 9.4 s and are executed for three levels of desired angular motor velocity, which are provided to the motor controller as desired set points: 40, 50 and 60 rad/sec. Note that, due to the non-ideal behavior of the system and the limitations of the PID controller, the desired set point results in practice in a varying angular velocity.

The measured and the estimated motor angle θ, rotational velocity $\dot{\theta }$, and the slider translational acceleration $\ddot{Y}$ are compared in Figure 6.

Three subsets of this full timespan are shown in Figure 7 to better appreciate the transient effects during the start-up and the two velocity transitions.

It is clear from Figure 6 and Figure 7 that the ADE-KF with the MB model tracks these various (derivative) states very well, underlining the well represented system kinematics.

In Figure 8 the estimated motor torque is compared to the measured motor torque.

This comparison shows a good prediction over the full time series (on the left) and on the angular velocity transitions (zoom-in figures on the right) thanks to the proposed methodology.

In Table 3 the root mean square error of the estimated virtual sensors and input torque are reported, underlying the relatively high accuracy of the estimated quantities. It is defined as $Error_{RMS}=\sqrt{\frac{\sum_{k}{\left[\chi_{m}\left(k\right)-\chi_{e}\left(k\right)\right]}^{2}}{n_{k}}}$, where $\chi_{m}$ and $\chi_{e}$ are the measured and estimated variables respectively, while $n_{k}$ is the total number of data samples.

The choice of performing the experiments for a relatively long timespan was made to demonstrate the filter stability in terms of both mean value and covariance prediction. For these kind of applications, where the uncertainty is dominated by difficult to model load effects (friction, etc.), the choice of focussing the model covariance on the input load allows effectively propagating the uncertainties. The dashed blue curves in Figure 8 represent the estimated expected variation of the augmented average state estimate within the 70% of confidence. It is expressed in terms of the standard deviations $\sigma_{d}=\sqrt{P_{d}^{+}}$ computed for each $k^{th}$ filter step where $P_{d}^{+}$ is the diagonal term of the a posteriori estimated covariance associated with the augmented state (disturbance). It is important to notice that the experimental motor torque (in red) remains bounded by the estimated input uncertainty therefore being an accurate estimation of the real covariance.

These results for the estimated torques show a significantly larger error than those obtained for the (derivative) states. For multibody problems in general, the state-estimates can be expected to be dominated by the kinematics of the system, which are generally well known. For the load estimates however, the dynamics of the system will play a crucial role. Besides key dynamic quantities such as the system inertia which can be modelled very accurately, other effects such as friction forces also influence this outcome. In this work we employed the friction model from Equation (Equation 56), but any error on this model is also propagated to the torque estimates. Due to the complex nature of the interface conditions for the slider and the rail (and the other bearings in the system), some errors are to be expected here. Key for future research will therefore be to examine how these complex load phenomena can be accounted for as effectively as possible. Moreover, by only employing one physical measurement ($\dot{\theta }$) in the ADE-KF scheme, it is guarantied that the accuracy of the estimated input torque would be less accurate for a poorly identified model. This can be observed in [15] where the same validation case was considered. Here, Angeli et al. proposed a deep learning methodology to reduce the initial MB equations from redundant to minimal coordinates where the resulting equations are then employed in an augmented extended Kalman filter scheme. Alternatively to what is presented in the current contribution, the supposed unknown motor torque is estimated employing the slider acceleration ($\ddot{Y}$) and no slider-track friction model was considered which has led to slightly less accurate input and virtual sensors estimation as compared to the currently proposed approach and model.

An important aspect in estimation problems is the choice of the Kalman filter parameters such as the process and measurement noise covarince matrices, ${\tilde{Q}}_{k}$ and ${\tilde{R}}_{k}$. It is recurrent while dealing with KF-based estimators that the accuracy of the estimated quantities is highly influenced by the selection of those parameters. However, general rules are not available since the filter parameters and influence strongly depend on the application case. Therefore, it is common to resort to a tuning step as the process of investigating and selecting these parameters. In the context of this work, the adopted strategy is described in the following section.

### 5.3. Kalman Filter Tuning

To attain the best accuracy from the presented estimation scheme, the model and measurement covariances need to be judiciously selected. In this work we have started with the selection of the measurement covariance matrix ${\tilde{R}}_{k}$ associated with the augmented measurement equations $\tilde{y}$. In lack of other information, We assume this measurement covariance constant over time. The covariance results from the combination of two main measurement contributions: the (motor) angular velocity $\dot{\theta }$ and the augmented constraint measurements ϕ, which reads as
(57)$${\tilde{R}}_{k}=\left[\begin{array}{cc} R_{\dot{\theta }} & 0_{\lambda }^{T}\\ 0_{\lambda } & R_{\phi } \end{array}\right],$$
where $R_{\dot{\theta }}$ is generally tuned based on reference noise measurements while for this application, since no noise reference was available, the author has chosen a value which is representative of the encoder measurement noise: $R_{\dot{\theta }}=10^{-2}$ (rad${}^{2}$/s${}^{2}$). Moreover, the authors have experienced that the influence of the measurement noise parameter $R_{\dot{\theta }}$ is relative to the value of the process noise covariance ${\tilde{Q}}_{k}$.

$R_{\phi }$ instead is linked to the mathematical and physical meaning of the constraint equations. In MB applications, we can distinguish two types of constraint equations: the ones that come from the inherent coordinates formulations, i.e., $\phi_{c}$ (e.g., Euler parameters should be unit vector and rotation matrix should be orthogonal), and the ones that come from joints definition, i.e., $\phi_{j}$, as for instance the spherical and/or revolute joints. In this work, for the latter a small noise term is allowed (i.e., all diagonal terms of $R_{\phi_{j}}$ are set to $10^{-9}$ while the off-diagonal terms are set to zero) representing the joint imperfections typical of real systems, whereas for the former, they are treated as perfect measurements (i.e., $R_{\phi_{c}}=0$), otherwise the kinematics and the mathematical principles that are used to describe the MB system are no longer valid.

Similarly to the augmented measurement covariance ${\tilde{R}}_{k}$, the augmented process noise matrix ${\tilde{Q}}_{k}$ is assumed constant for all filter steps and it can be written as combination of the system and augmented states contributions as
(58)$${\tilde{Q}}_{k}=\left[\begin{array}{cc} Q_{x} & 0\\ 0 & Q_{d} \end{array}\right].$$

As we assume the model to be practically perfect compared to the high uncertainty on the unknown inputs, the process noise matrix $Q_{x}$ associated with the system states is set to zero.

The remaining parameter $Q_{d}$ associated with the unknown input torque (disturbance) is obtained in a brute-force optimization fashion. Since in this case there is only a single value to be chosen, an exhaustive search is therefore not computationally prohibitive. In this regard a grid of $Q_{d}$ values have been selected, going from $10^{-5}$ to $10^{5}$ sampled exponentially in eleven increments, leading to a corresponding eleven performed estimations. The choice of the $Q_{d}$ is based on the L-curve method proposed by [28] where its nearly optimal value (Not optimal in absolute sense due to discrete evaluation points $Q_{d}$ and the user defined $R_{\dot{\theta }}$) is such that the best trade-off is achieved between the *Error norm* and *Smoothing error*. These norms are defined as
(59)$$\begin{array}{c} Error\;norm=\sum_{k=1}^{n_{k}}\left|\right|y_{k}^{*}-{\tilde{y}}_{k}^{-}{\left|\right|}_{2} \end{array}$$
and,
(60)$$\begin{array}{c} Smoothing\;norm=\frac{\sum_{k=1}^{n_{k}}{\left||d\left(k\right)|\right|}_{2}}{\sum_{k=1}^{n_{k}}{\left||T\left(k\right)|\right|}_{2}} \end{array}$$
where d(k) and T(k) are the estimated (disturbance) and measured torque respectively, while $n_{k}$ is the total number of sampling point *k*. On the left side of Figure 9 the L-curve for the evaluated $Q_{d}$ grid points is shown where the blue marked point is the chosen value corresponding to $Q_{d}=10^{0}$. This value have been used for the results shown in Figure 6, Figure 7 and Figure 8. As expected, it is observed that the error norm decreases with an increasing smoothing error which corresponds to an increasing $Q_{d}$ till a saturation is reached. This occurs when a further increase of process noise value does not show significant improvements to the estimation results ($Q_{d}>10^{0}$).

On the right side of Figure 9 the estimated torque with the chosen $Q_{d}$ (marked in blue) is compared to a sub-optimal value (marked in orange) on the velocity transitions zones while in Figure 10 the full time series are compared.

In Figure 9 and Figure 10, it can be seen that despite a relatively similar tracking of the input (more delayed), at low rotational speed using a smaller covariance than the “optimal” one, the estimation accuracy degrades over time with a worse tracking of estimated torque.

Although this study does not address the full scope of noise covariance tuning, the author deemed it sufficient to explain the methodological developments considered in this contribution. Further research on covariance tuning might be necessary to obtain a more holistic approach.

## 6. Discussion and Conclusions

This work presents a new estimation methodology tailored for MB models to enable the definition of virtual sensors for various system states and inputs.

Through the choice of a general MB modeling approach various key physical effects can be accurately accounted for, ranging from nonlinear kinematics to complex dynamic effects. The developed framework allows using these multibody models in an estimation framework without particular additional modeling assumptions or reformulations. More specifically, no constraint elimination methods are required to employ the defined MB model into the estimation framework, reducing the preparation time and the user effort to setup the estimation problem while ensuring the correct physical and mathematical interpretation of the system under investigation. As the proposed methodology has no particular assumptions with respect to the multibody model formulation it can be easily extended to any of the commonly available (flexible) MB approaches, e.g., FNCF, FFR-CMS, or GCMS. However, to fully benefit from the proposed approach the equations of motion and tangent stiffness matrices of the system should be analytically available to efficiently assemble the linearized system and measurement matrices.

In the present work, we exploited the FNCF MB approach, as this methodology inherently enables an easy and efficient evaluation of the different tangent model matrices required for the estimation framework.

Finally, the developed methodology has been experimentally validated on a slider-crank mechanism. Very high accuracy is obtained for the estimated states with respect to the available measurements. Good accuracy is also obtained for the estimated input torque, but due to the larger dynamic model errors in e.g., friction effects the resulting errors are higher than those obtained for the states. The validation has been performed over a relatively large time-span which also demonstrates the capability of the presented framework to obtain long-term stable estimates with a bounded uncertainty, in the form of a bounded covariance.

The presented methodology has some drawbacks since a large number of states (including the Lagrange multipliers) and measurements (including the constraints equations) are employed. These lead to a computationally less efficient approach as compared to other state of the art techniques (e.g., using minimal coordinates [15]). A possible solution to mitigate this issue might come from a wise selection of the number of bodies and constraints equations to construct the high-fidelity model. For instance, in this contribution, the choice of redundant number of bodies and constraints was made in purpose to stress the potential of the proposed methodology while dealing with redundant set of DAEs equations. More precisely, not all bodies and hence constraints were required to achieve the estimated motor torque with the same level of accuracy (e.g., the motor housing and the different supports). Nevertheless, if computational efficiency is not a limiting factor, i.e., if an online estimation is not required, this approach has the potential to enable a very generic deployment of (flexible) multibody-based state-input estimation.

Future work will focus on how these methodologies can be employed to obtain more accurate descriptions for key dynamic effects such as the friction present in these multibody systems.

## Figures and Tables

**Figure 1 sensors-21-04495-f001:**
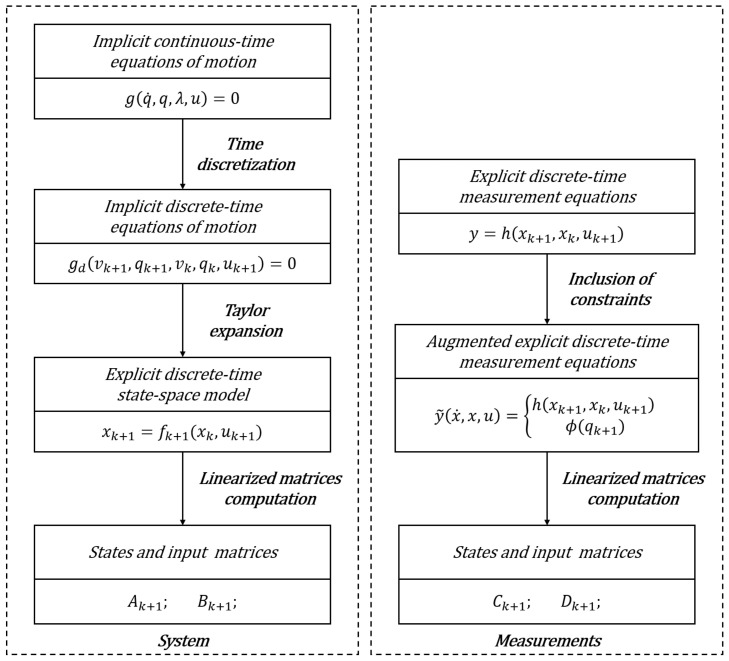
Block diagram representation of the system and measurement matrices computation for a generic integration step.

**Figure 2 sensors-21-04495-f002:**
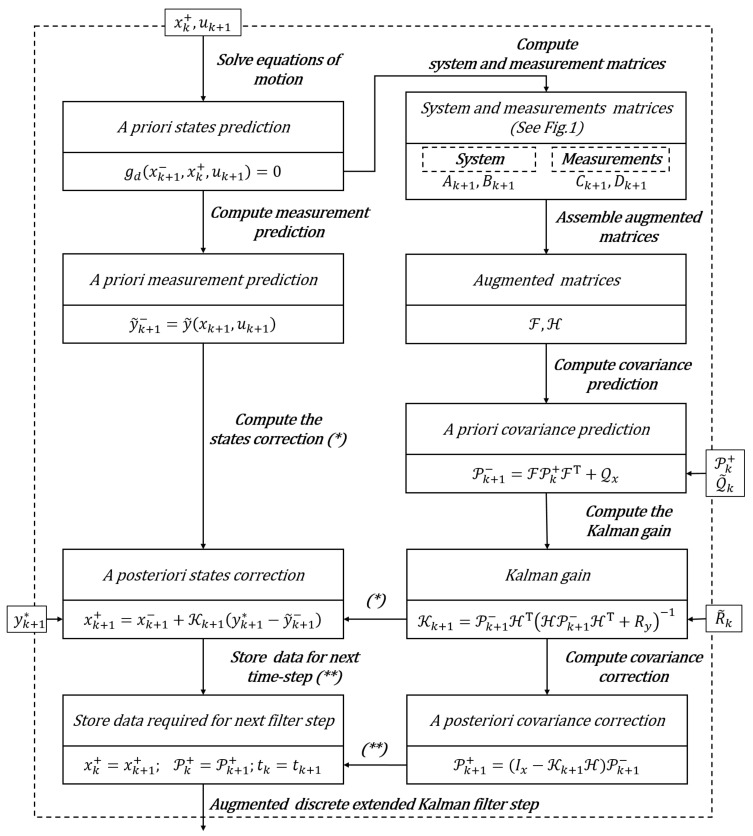
Block diagram representation of the recursive ADE-KF scheme for a generic filter step.

**Figure 3 sensors-21-04495-f003:**
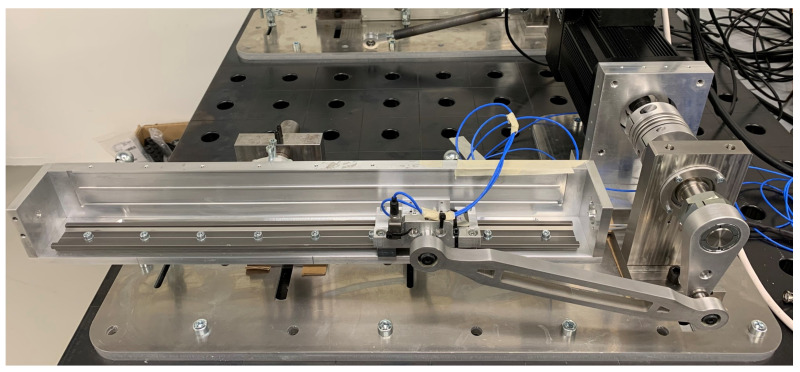
The slider-crank: experimental setup.

**Figure 4 sensors-21-04495-f004:**
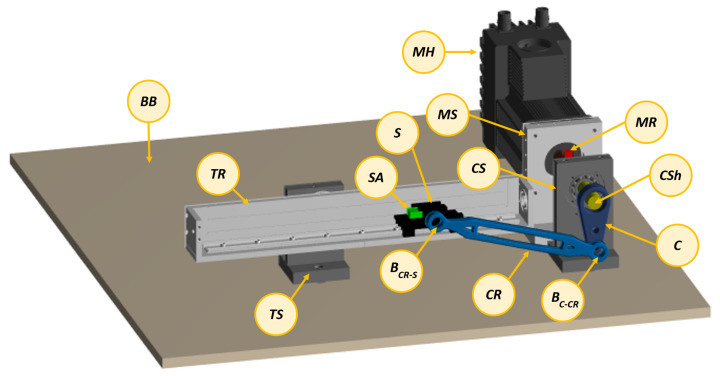
The slider-crank: multibody model.

**Figure 5 sensors-21-04495-f005:**
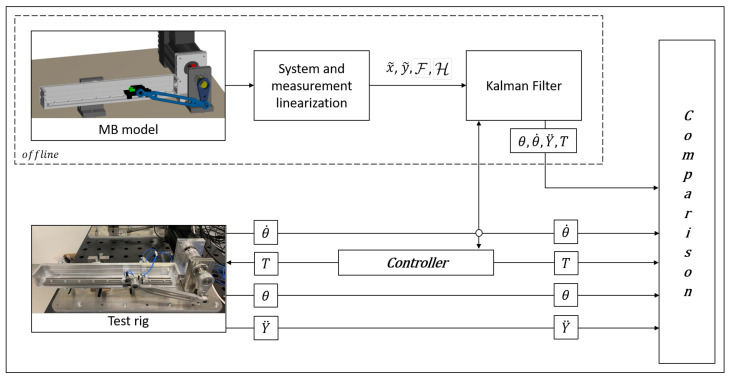
Diagram of the coupled state-input estimation scheme and signal comparisons. θ and $\dot{\theta }$ are the motor angle and angular motor velocity respectively; $\ddot{Y}$ is the translational slider acceleration; *T* is the motor torque.

**Figure 6 sensors-21-04495-f006:**
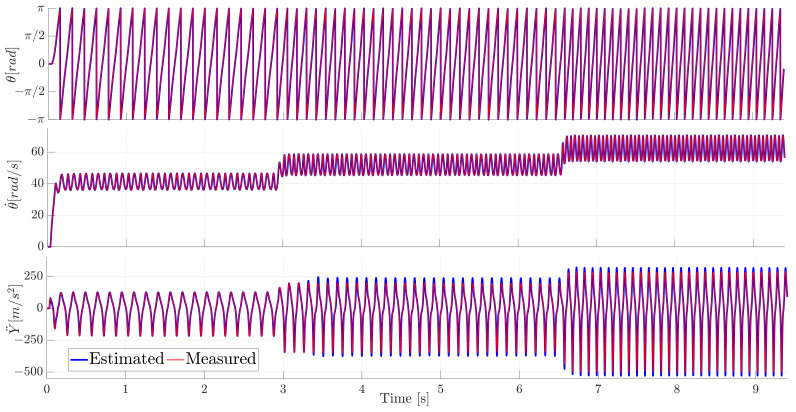
Comparison of the measured and estimated motor angle θ (**top**), motor angular velocity $\dot{\theta }$ (**middle**) and translational slider acceleration $\ddot{Y}$ (**bottom**) for the full time series.

**Figure 7 sensors-21-04495-f007:**
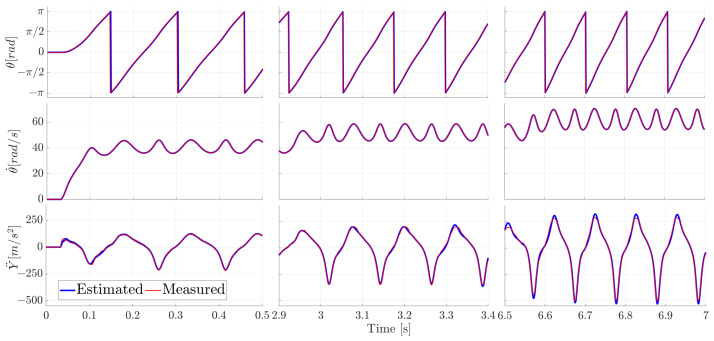
Comparison of the measured and estimated motor angle θ (**top row**), motor angular velocity $\dot{\theta }$ (**middle row**) and translational slider acceleration $\ddot{Y}$ (**bottom row**). Zoom-in per column on the velocity transitions.

**Figure 8 sensors-21-04495-f008:**
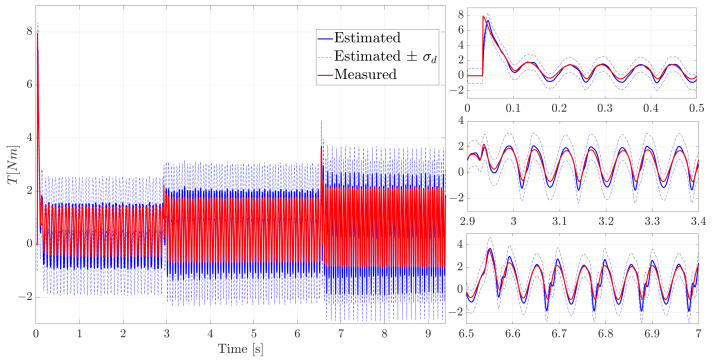
Comparison of the measured and estimated motor torque; on the left, full time series; on the right, the zoom-in on the velocity transitions are shown.

**Figure 9 sensors-21-04495-f009:**
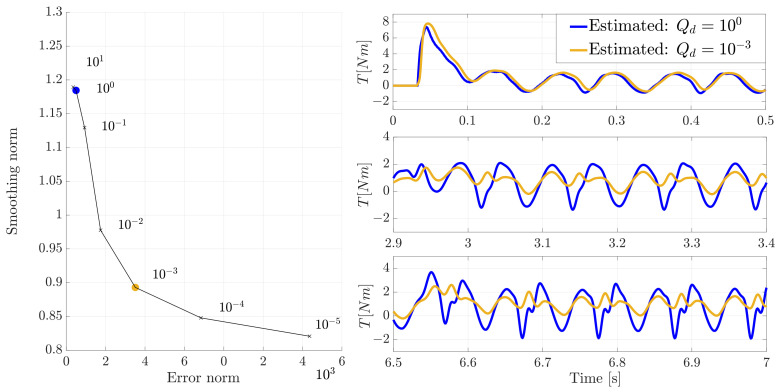
The L-curve plot for different process variance $Q_{d}$ (**left figure**) and zoom-in comparison on the velocity transition of the measured and estimated motor torque using two different values of process variance $Q_{d}$ (**right figure**).

**Figure 10 sensors-21-04495-f010:**
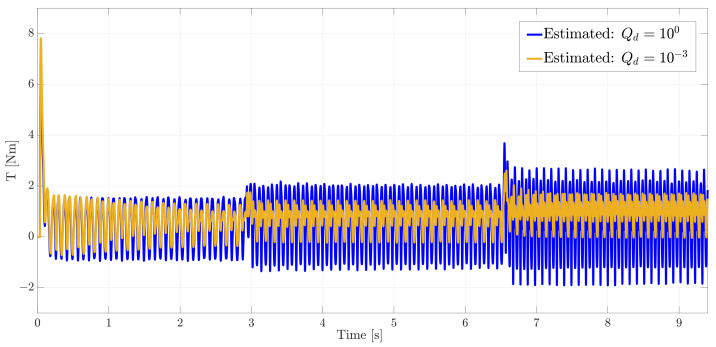
Comparison of the estimated motor torque using two different values of process covariance $Q_{d}$ of the augmented state.

**Table 1 sensors-21-04495-t001:** Mechanical properties expressed with respect to the center of gravity of each individual body.

Body	*m* [kg]	$J_{\mathrm{xx}}$ [kg · m${}^{2}$]	$J_{\mathrm{yy}}$ [kg · m${}^{2}$]	$J_{\mathrm{zz}}$ [kg · m${}^{2}$]	
BB	$3.2175\times 10^{3}$	$2.9225\times 10^{2}$	$6.274\times 10^{2}$	$8.714\times 10^{2}$
MH	$1.420\times 10^{1}$	$2.48647\times 10^{-2}$	$9.366\times 10^{-2}$	$9.366\times 10^{-2}$
MR	$6.670\times 10^{-1}$	$2.947\times 10^{-3}$	$2.001\times 10^{-3}$	$2.001\times 10^{-3}$
MS	1.350	$6.052\times 10^{-3}$	$3.897\times 10^{-3}$	$2.346\times 10^{-3}$
*C*	$1.830\times 10^{-1}$	$5.742\times 10^{-4}$	$4.930\times 10^{-4}$	$1.522\times 10^{-5}$
CSh	$4.960\times 10^{-1}$	$1.644\times 10^{-4}$	$7.068\times 10^{-4}$	$7.068\times 10^{-4}$
CS	$8.4058\times 10^{-1}$	$2.678\times 10^{-3}$	$1.954\times 10^{-3}$	$8.047\times 10^{-4}$
$B_{C-CR}$	$2.820\times 10^{-1}$	-	-	-
CR	$2.540\times 10^{-1}$	$1.185\times 10^{-2}$	$1.1840\times 10^{-2}$	$3.654\times 10^{-5}$
$B_{CR-S}$	$2.820\times 10^{-1}$	-	-	-
*S*	$2.562\times 10^{-1}$	$2.665\times 10^{-4}$	$1.131\times 10^{-4}$	$3.29510^{-4}$
SA	$2.200\times 10^{-2}$	-	-	-
TR	1.206	$2.800\times 10^{-1}$	$9.424\times 10^{-3}$	$2.800\times 10^{-1}$
TS	4.206	$1.392\times 10^{-2}$	$1.256\times 10^{-2}$	$4.963\times 10^{-3}$

**Table 2 sensors-21-04495-t002:** Pacejka friction model parameters.

$k_{c}$ [N/m]	$c_{c}$ [Ns/m]	b [s/m]	c[−]	d[−]	e[−]
$9.7854\times 10^{6}$	1.196	$5.036\times 10^{2}$	1.5708	$2.653\times 10^{-2}$	−9.8534

**Table 3 sensors-21-04495-t003:** Accuracy of the estimated quantities in terms of root mean square error.

	θ [rad]	$\dot{\theta }$ [rad/s]	$\ddot{Y}$ [m/s${}^{2}$]	*T* [Nm]
$Error_{RMS}$	0.005	0.051	13.651	0.334

## Data Availability

The data presented within this study are resulting from activities within the acknowledged projects and are available therein.

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
