# Peer review of "A Discrete-Time Extended Kalman Filter Approach Tailored for Multibody Models: State-Input Estimation"

_sensors, 2021, doi:10.3390/s21134495_

Round 1

Reviewer 1 Report

The authors did a good job, however, I have some recommendations.

  1. Please improve your introduction with more information, there are several papers with information that can improve it.
  2. Please improve the legend of figure 6.
  3. In line 130 there is an error.
  4. In abbreviations, there are some errors.

Author Response

The authors did a good job, however, I have some recommendations.

Dear reviewer, thank you so much for the suggested improvements and appreciation of the presented work.

Point 1: Please improve your introduction with more information, there are several papers with information that can improve it.

Response 1: I’ve included additional information in the introduction as indicated.

Point 2: Please improve the legend of figure 6.

Response 2: I’ve modified the legend of figure 6 as indicated.

Point 3: In line 130 there is an error.

Response 3: I’ve corrected the error in line 130.

Point 4: In abbreviations, there are some errors.

Response 4: I’ve checked and corrected the errors.

Reviewer 2 Report

The paper deals with a problem of unknown input estimation for a class of mechanical multi-body systems.

The main contribution is in the formulation of the estimation framework providing a systematic means of derivation of the prediction model. This is achieved by transforming a set of differential-algebraic equations of motion to a standard state-space form for which a standard Kalman filtering approach may be applied.

The paper is well structured and easy to follow. The results are highly relevant for experiments with mechanical systems with a need for indirect estimation of an unmeasurable quantity.

My only concern related to the clarity of the presentation is the classical issue of the Kalman filter – how to tune the knobs available (the state/measurement (co)variances) to achieve reasonable performance? In your particular case, the estimation results will depend on the chosen variance of the unknown input disturbance. High values will allow faster tracking at the cost of propagation of the measurement noise and vice versa. While it may be quite simple to tune the filter with the “unknown” quantity available from measurement for reference, it is a bit of a headache in a practical scenario where it is really unavailable.

Please consider adding:

  1. Discussion of this topic that is of crucial practical importance
  2. The procedure of the covariance adjustment that you used for setting the filter in your slider-crank scenario, perhaps complemented by a study of the influence of the unknown input variance setting on the estimation error

Apart from this issue, the paper seems to be ready for publication in my opinion.

Author Response

Dear reviewer, first of all thank you so much for the your punctual review.

Point 1: Discussion of this topic that is of crucial practical importance.

Response 1: I agree that filter tuning is troublesome in practical scenarios. Although, to my knowledge there are no generic rules that can be applied for each specific application to deal with this issue. In the revised version of the paper I’ve included the followed Kalman filter tuning approach for the considered application.

Point 2: The procedure of the covariance adjustment that you used for setting the filter in your slider-crank scenario, perhaps complemented by a study of the influence of the unknown input variance setting on the estimation error.

Response 2: I’ve included an additional study on the influence of the process noise covariance on the estimation error to the new version of the paper.

Apart from this issue, the paper seems to be ready for publication in my opinion.

Reviewer 3 Report

This paper proposed combining high-fidelity multibody models with a Kalman filter-based estimator to estimate state-input unknown forces. The proposed methodology is validated experimentally validated on a slider-crank mechanism, focusing on estimating the input torque delivered by a driver electro-motor together with the system states. Overall this paper is well prepared, and the results are intriguing. Below are some comments to be considered:

  • A flow chart of the proposed work would be helpful
  • Check on missing symbols, such as line 130.
  • Line 373. If it is assumed that all model uncertainties is dominated by the augmented state, representing the unknown input, by setting Qx = 0? What is the purpose of having random walk as the unknown inputs? And why random walk and not other algorithms?
  • Figure 4-6 can be enlarged. Figure 6, the current horizontal direction is better presented as the vertical direction, and Cyan color for the line is hard to see.
  • Please discuss potential drawbacks in the proposed method during implementation and how to overcome it.

Author Response

Dear reviewer, first of all, thank you so much for your review giving me the possibility to improve the quality of the contribution.

Point 1: A flow chart of the proposed work would be helpful.

Response 1: In the revised version of the paper I’ve included a flowchart of the methodological approach as indicated.

Point 2: Check on missing symbols, such as line 130.

Response 2: I’ve included the symbol of line 130 and additionally checked on missing ones.

Point 3: Line 373. If it is assumed that all model uncertainties is dominated by the augmented state, representing the unknown input, by setting Qx = 0? What is the purpose of having random walk as the unknown inputs? And why random walk and not other algorithms?

Response 3: In line 373 indeed I meant Qx = 0 and more elaborate guidelines have been introduced in the new version of the paper.  The random walk model is chosen in this work since it is assumed that no prior information on the input is known which is the case for many general applications. On the other hand, if information on the input is available (e.g. periodic input) other input models can definitely be included to improve its estimation.

This has been added to the text.

Point 4: Figure 4-6 can be enlarged. Figure 6, the current horizontal direction is better presented as the vertical direction, and Cyan color for the line is hard to see.

Response 4: The figures have been modified according to your review as well as to accommodate other reviews.

Point 5: Please discuss potential drawbacks in the proposed method during implementation and how to overcome it.

Response 5: The main drawbacks of the methodology have been addressed in the new version of the paper.

Reviewer 4 Report

I found this paper very interesting and really enjoyed reviewing it. The paper is written in good English and presents important outcomes. The TurnItIn antiplagiarism check showed only 17% of paper similarity, which is a very good index value. However, there are few issues that should be explained more clearly. Thus, I recommend a minor review.

1. The authors should add a block diagram of the algorithm in order to explain their approach in a more structured way.

2. The authors should assess the estimation results also by using conventional validation criteria as e.g. in [R1]. This is required as Kalman Filter tends to provide a good visual match even for poorly performed identification (e.g. in overstructured models). If the authors cannot provide the numerical values, this should be discussed at least.
[R1] R.V. Jategaonkar, Flight Vehicle System Identification: A Time Domain Methodology, AIAA, Reston, VA, 2006

3. Please provide the (quantitative) information about the standard deviation shown in Figure 6.

I hope that the authors will take all my comments into consideration.

Author Response

Dear reviewer, thank you so much for your interesting comments and suggestions.

Point 1: The authors should add a block diagram of the algorithm in order to explain their approach in a more structured way.

Response 1: I’ve included the block diagram of the methodology as suggested for a more structured contribution.

Point 2: The authors should assess the estimation results also by using conventional validation criteria as e.g. in [R1]. This is required as Kalman Filter tends to provide a good visual match even for poorly performed identification (e.g. in overstructured models). If the authors cannot provide the numerical values, this should be discussed at least.

[R1] R.V. Jategaonkar, Flight Vehicle System Identification: A Time Domain Methodology, AIAA, Reston, VA, 2006

Response 2: In the revised version of the paper I’ve included the RMS error of each estimated quantity and briefly discussed the problem of poorly identified models.

Point 3: Please provide the (quantitative) information about the standard deviation shown in Figure 6.

Response 3: The standard deviation indicated in Figure 6 stands for the estimated augmented state variance of the a posteriori state covariance matrix. In the revised version of the paper this concept is clarified and more details are given.